# Internal Strain Distribution of Laser Lap Joints in Steel under Loading Studied by High-Energy Synchrotron Radiation X-rays

**Takahisa Shobu [1,\*], Ayumi Shiro [2], Fumiaki Kono [3,4], Toshiharu Muramatsu [5], Tomonori Yamada [5,6], Masayuki Naganuma [7] and Takayuki Ozawa [8]**

1   Materials Sciences Research Center, Japan Atomic Energy Agency, 2-4 Shirakata, Tokai-mura, Naka-gun, Ibaraki 319-1195, Japan

2   Synchrotron Radiation Research Center, National Institutes for Quantum and Radiological Science and Technology, 1-1-1 Kouto, Sayo-cho, Sayo-gun, Hyogo 679-5148, Japan; shiro.ayumi@qst.go.jp

3   Advanced Nuclear System Research and Development Directorate, Japan Atomic Energy Agency, 4-33 Muramatsu, Tokai-mura, Naka-gun, Ibaraki 319-1194, Japan; kono.fumiaki@qst.go.jp

4   Institute for Quantum Life Science, National Institutes for Quantum and Radiological Science and Technology, 2-4 Shirakata, Tokai-mura, Naka-gun, Ibaraki 319-1106, Japan

5   Tsuruga Comprehensive Research and Development Center, Japan Atomic Energy Agency, 65-20 Kizaki, Tsuruga, Fukui 914-8585, Japan; muramatsu.toshiharu@jaea.go.jp (T.M.); tyamada@werc.or.jp (T.Y.)

6   Research & Development Department, The Wakasa Wan Energy Research Center, 64-52-1 Nagatani, Tsuruga, Fukui 914-0192, Japan

7   Nuclear Fuel Cycle Engineering Laboratories, Japan Atomic Energy Agency, 4-33 Muramatsu, Tokai-mura, Naka-gun, Ibaraki 319-1194, Japan; naganuma.masayuki@jaea.go.jp

8   Fuel Cycle Design Department, Japan Atomic Energy Agency, 4002 Narita-cho, Oarai, Higashiibaraki-gun, Ibaraki 311-1393, Japan; ozawa.takayuki@jaea.go.jp

\*   Correspondence: shobu@spring8.or.jp; Tel.: +81-791-58-0308

**Abstract:** The automotive industries employ laser beam welding because it realizes a high energy density without generating irradiation marks on the opposite side of the irradiated surface. Typical measurement techniques such as strain gauges and tube X-rays cannot assess the localized strain at a joint weld. Herein high-energy synchrotron radiation X-ray diffraction was used to study the internal strain distribution of laser lap joint PNC-FMS steels (2- and 5-mm thick) under loading at a high temperature. As the tensile load increased, the local tensile and compressive strains increased near the interface. These changes agreed well with the finite element analysis results. However, it is essential to complementarily utilize internal defect observations by X-ray transmission imaging because the results depend on the defects generated by laser processing.

**Keywords:** high energy synchrotron radiation; internal strain distribution; laser lap welded steel; in situ measurement

## 1. Introduction

Laser processing technology realizes a higher energy density than other welding heat sources. As laser processing technology results in welding due to the focal spot, this technology produces welding with less thermal strain. Lap joint welds with two thin plates can be joined without generating irradiation marks on the opposite side of the laser irradiation surface. Consequently, the automotive and other industries have adopted laser beam welding. The Japan Atomic Energy Agency (JAEA) has been conducting research and development on laser lap joint welding as a technology for ferrite/martensitic steel (PNC-FMS) with excellent swelling resistance. PNC-FMS can increase the burnup of fuel in fast reactors and next-generation nuclear reactors.

Traditionally, stress measurements using strain gauges and tube X-rays are performed to assess stress distribution. However, these techniques only provide the stress information

about the surface or the overall average. It is difficult to estimate the internal stress/strain state from the surface measurement results under a load when the target is lap welded joints, especially since a laser should generate a stress gradient in the local region of the welded portion. A method using neutrons can realize nondestructive measurements inside a material. Although this method has been applied to welded portion measurements [1–4], it is rarely used to measure minute parts.

High-energy X-rays have an extremely high penetrating power for materials and can measure thicknesses of the order of millimeters, even for steel materials. This study used high-energy synchrotron radiation X-rays to measure the internal strain distribution of laser lap joint PNC-FMS steels (2 and 5 mm thick) under loading at a high temperature. In addition, the validity of this measurement method was examined.

## 2. Materials and Methods

PNC-FMS, which holds promise for the internal ducts of next-generation nuclear reactors, was used as the test material. Table 1 shows the chemical components. Two specimens with dimensions of 70 × 6 × 2 and 70 × 6 × 5 mm were overlapped, and an Yb fiber laser with a power of 1.3 kW and a heat input of 1500 J was irradiated and joined them from the 2-mm thick plate side. To remove strain due to laser processing, the specimen was heat treated at 690 °C for 103 min [5]. Figure 1a,b show the top and side views of the specimen, respectively. The specimen was welded with plates on both sides so that a tensile stress could be easily applied and placed on the mounting jig of the high-temperature load device shown in the next section. Figure 2 shows a cross-section metallographic image of the post-experiment specimen around the weld. The melting zone and heat affected zone (HAZ) are clearly discernible, confirming that laser welding reached the lower plate, and the effect was about 1.5-mm in thickness.

**Table 1.** Chemical composition of PNC-FMS (target composition).

| Element | C | Si | Mn | Ni | Cr | Mo | W | V | Nb | N |
|---|---|---|---|---|---|---|---|---|---|---|
| Composition (wt%) | 0.12 | 0.05 | 0.60 | 0.40 | 11.0 | 0.5 | 2.0 | 0.20 | 0.05 | 0.05 |

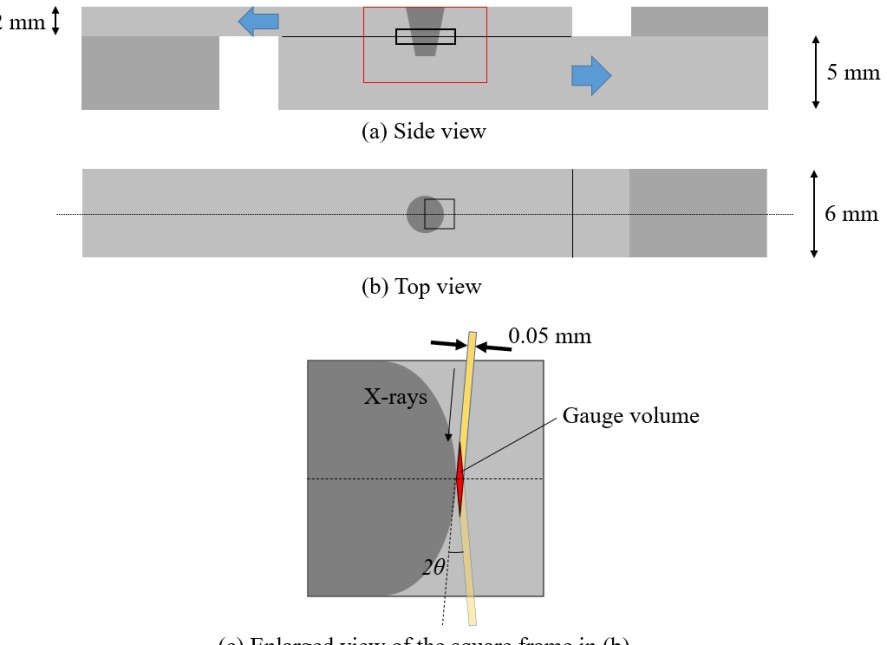

(a) Side view

(b) Top view

(c) Enlarged view of the square frame in (b)

**Figure 1.** Specimen.

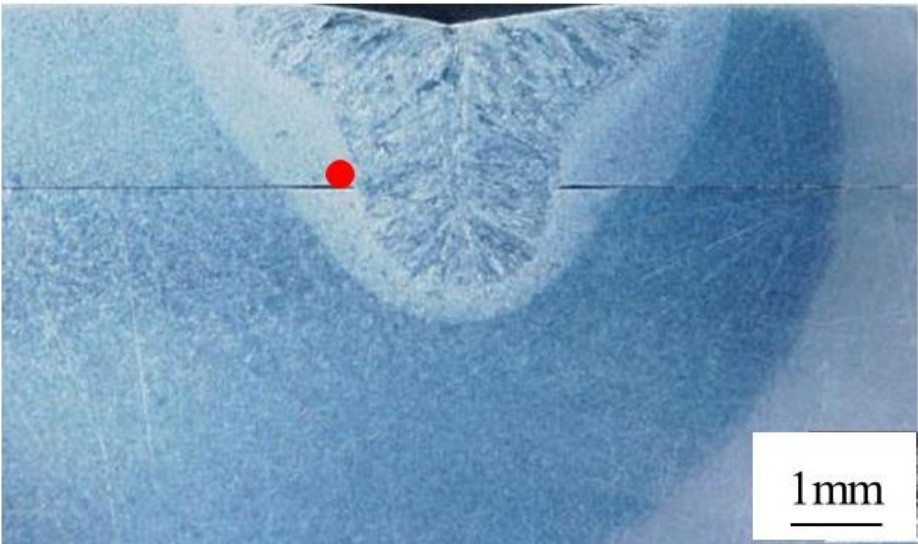

**Figure 2.** Cross-section metallographic image after the experiment.

We performed two experiments at BL22XU in SPring-8 [6]. A regression analysis using multiple diffraction planes obtained from $CeO_2$ powder indicated that the X-ray energy was 70.11 keV. An experiment was performed using a horizontal rotation type 2-axis diffractometer as a measuring device and a high-temperature load device installed at the center. Figure 3 shows the experimental setup. The slit on the incident side, which measured 0.05 mm in the diffraction angle direction and 0.15 mm in the vertical direction, adjusted the size of the synchrotron radiation X-ray irradiated onto the sample. A collimator and a slit, which were 0.05 mm in the diffraction angle direction and 5 mm in the vertical direction, were set up on the receiving side, and the diffracted X-rays from the local region inside the sample were measured by a cadmium telluride detector (XR-100T-CdTe, Amptek, Bedford, MA, USA). This setup allowed the diffraction profile from a local region (gauge volume) of the sample to be measured (Figure 1c). The internal strain distribution of the sample was obtained by moving the sample vertically and horizontally. The $\alpha$-Fe211 diffraction with a diffraction angle of about 8.56 deg was measured, and the strain was calculated using the following equation

$$\varepsilon = \frac{\Delta d}{d_0} = \frac{d - d_0}{d_0} = -\frac{sin\theta - sin\theta_0}{sin\theta} \tag{1}$$

Here, $d$ and $\theta$ are the lattice spacing and the diffraction angle measured in the local region of the specimen, whereas $d_0$ and $\theta_0$ are the lattice spacing and the diffraction angle of the heat-treated and strain-relieved specimen, respectively.

Using the operational temperature inside a next-generation reactor as the standard condition, the two-dimensional strain distribution inside the black frame in Figure 1a was measured while heating at 530 °C and tensile loading was applied in the direction of the arrow in Figure 1a.

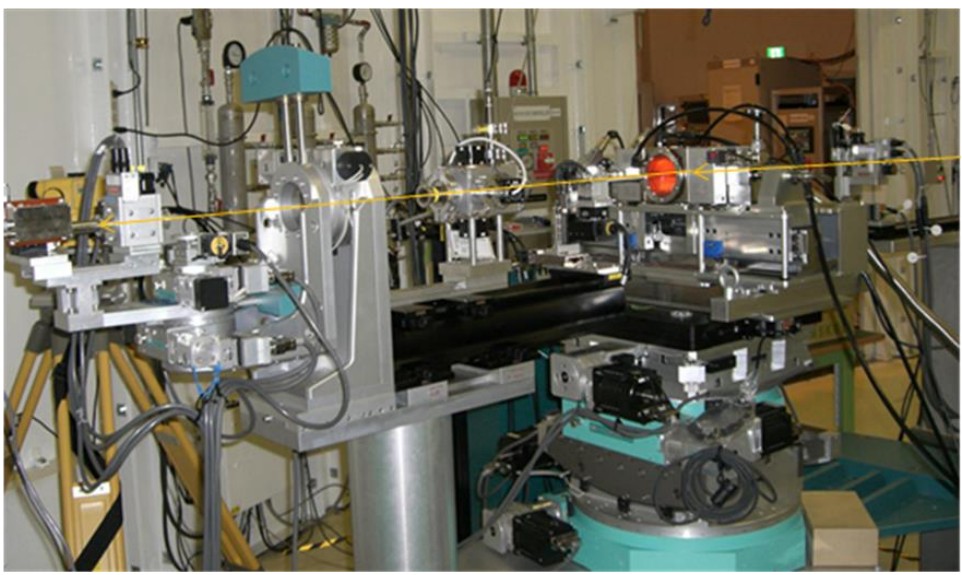

**Figure 3.** Photograph of the experimental setup.

## 3. Results

Figure 4 shows the α-Fe211 diffraction profile at the locations marked with a red dot in Figure 2 under a high temperature and tensile load. Diffraction intensity changed under loading. It is thought that the grains rotated and the number of diffracted grains accidentally increased at this measurement position, since the diffraction intensity changed randomly, regardless of the measurement position under loading. The diffraction profile gradually shifted to the lower angle side as the tensile load was applied to the specimen, because the tensile strain was locally applied to the measured part. Additionally, elastic deformation should have occurred since the diffraction profile width hardly changed with the tensile load.

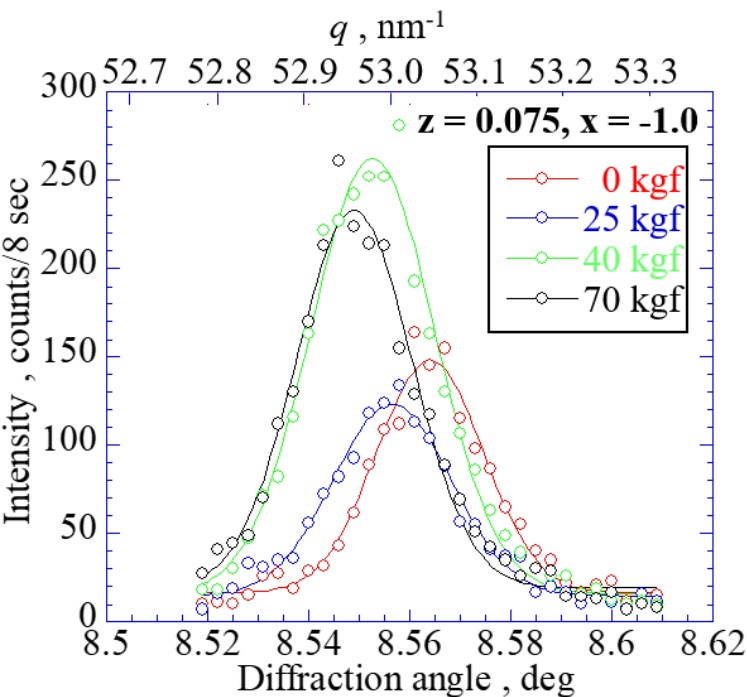

**Figure 4.** Diffraction profiles of α-Fe211 under a high temperature and tensile load.

Figure 5 shows the two-dimensional distribution of the full width at the half maximum (FWHM) of the $\alpha$-Fe211 diffraction profile and lattice strain at 530 °C under a tensile load. The trapezoid represents the fusion zone. The FWHM values in the trapezoid were larger, but they hardly changed with the load. According to the Vickers hardness measurements using the other specimens prepared under the same conditions, the hardness of the welded part before the heat treatment was about 1.7 times that of the base metal, but after the heat treatment, it was about 1.1 times [5]. This difference was attributed to the small amount of plastic strain remaining inside, even after the heat treatment. On the other hand, as for the strain shown in Figure 5d–f, tensile strain was observed at the interface under a tensile load, and a compressive strain was observed at the interface on the opposite side. Additionally, the strain increased with the load.

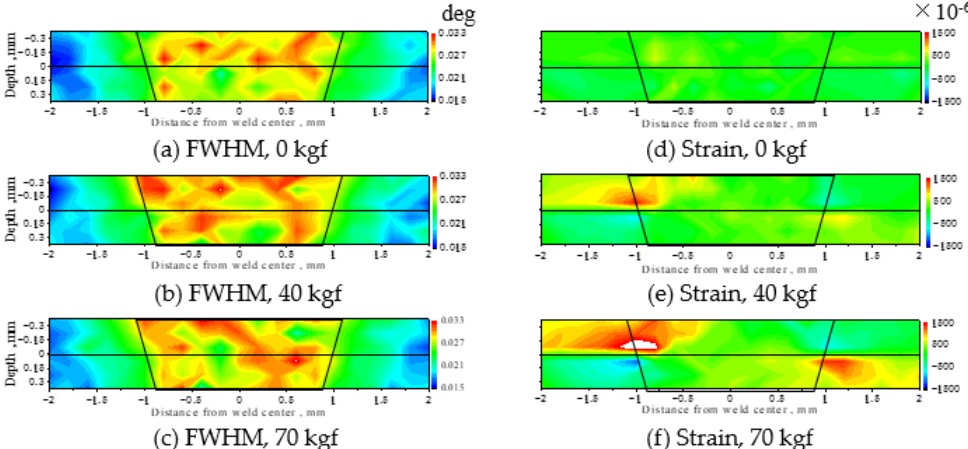

**Figure 5.** Two-dimensional distribution of the full width at the half maximum (FWHM) of the $\alpha-$Fe211 diffraction profile and lattice strain at 530 °C under a tensile load.

## 4. Discussion

To validate the measurement results, finite element method (FEM) analyses were performed using a finite element nonlinear structural analysis system, FINAS [7]. FINAS is a general-purpose nonlinear structural analysis system based on the finite element method, which was developed by JAEA. Figure 6 shows the analysis model, which modeled half of the specimen. The penetration diameter was set to 2 mm, based on the cross-sectional observation, and the gap between metal plates was set to 0.03 mm. The Young's modulus and Poisson's ratio of the PNC-FMS at 530 °C were 177 GPa and 0.32, respectively [8]. Although the physical properties of the weld must be considered, this analysis used the same physical property values as the base metal for simplicity. In this experiment, tensile loads in opposite directions were applied to the upper and lower plates. However, the constraint conditions were constant in the FEM analyses and the boundary conditions were determined using the load direction as a parameter.

Figure 7 shows the FEM analysis under a tensile loading of about 70 kgf. An extremely high tensile and compressive strain was observed in the limited part near the interface of the plates. In particular, a strong tensile strain of about $3000 \times 10^{-6}$ was obtained at the edge of the interface. This calculated value was qualitatively consistent with the experimental results in Figure 5f.

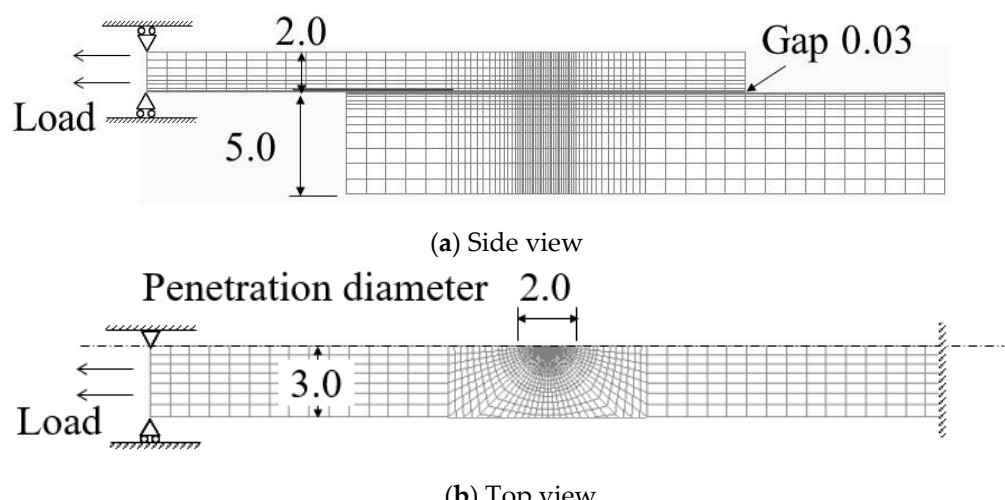

**Figure 6.** Size and FEM analysis model of the specimen. (**a**) Side view; (**b**) Top view.

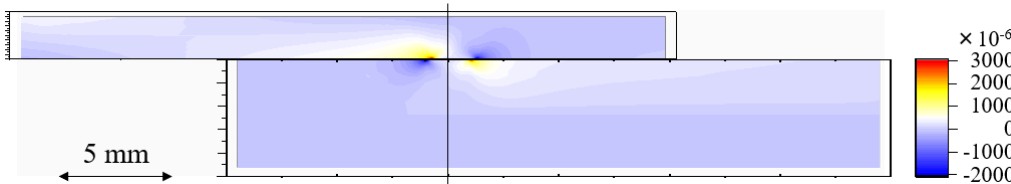

**Figure 7.** Size and FEM analysis model of the specimen.

Figure 8 shows the strain distribution near the interface from FEM and experimental measurements. The dotted and solid lines indicate the results of the measurements and FEM, respectively. The measured and FEM results agreed qualitatively. For example, a change in the tensile or compressive strain altered the strain in the weld. However, the change slightly differed quantitatively. The measured strain left of center was larger than the theoretical strain, whereas that on the right was smaller than the theoretical strain. It is possible that the tensile direction was tilted as a factor of this symmetrical difference. The load direction was generated horizontally on the left and right in the FEM because the load was small, but the load direction should have changed clockwise with an increased load.

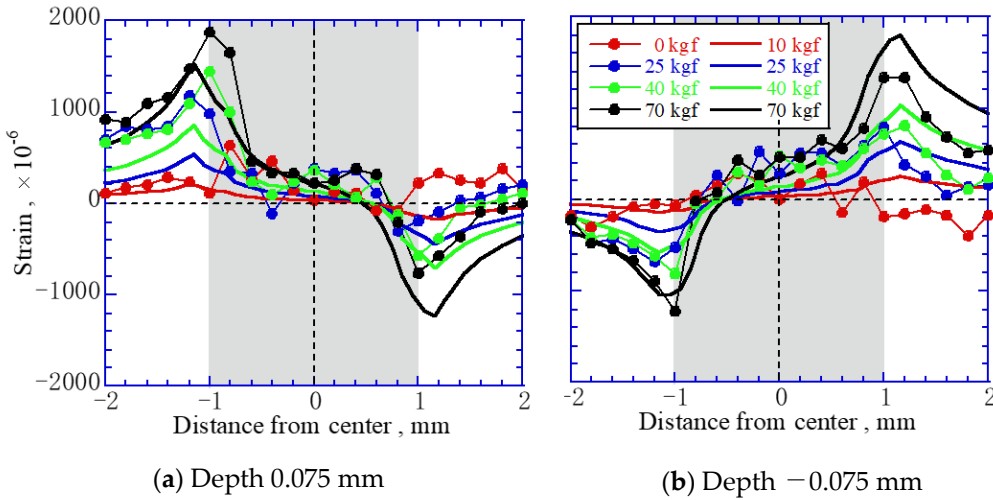

**Figure 8.** Strain distribution near the interface by FEM (solid line) and measurements (dotted line).

In this calculation, the Young's modulus of the welded part was the same as that of the base metal, which is the Young's modulus of PNC-FMS steel at 530 °C. On the other hand, a structural difference was observed between the welded part and the base metal (Figures 2 and 5). Their Vicker's hardness slightly differed. In this case, the physical property values such as the Young's modulus should also have differed slightly. Therefore, FEM analyses were performed by increasing the Young's modulus of the welded part by 1.1 times that of the base metal.

Figure 9 shows the strain distribution near the interface recalculated using the FEM analyses. As a result of changing the Young's modulus, the strain at –1 mm from the center increased, while that at 1 mm from the center decreased. These FEM analyses agreed well with the experimental results, confirming the effectiveness of adjusting the Young's modulus for each structure.

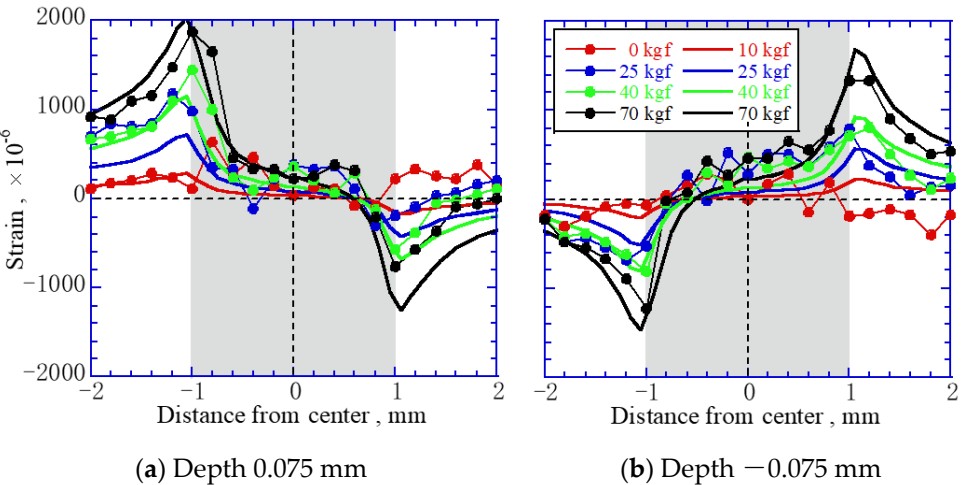

(**a**) Depth 0.075 mm        (**b**) Depth −0.075 mm

**Figure 9.** Strain distribution near the interface by FEM (solid line) and measurements (dotted line). FEM analyses were calculated assuming the Young's modulus of the weld was 1.1 times that of the base metal.

On the other hand, excessive temperature changes induced internal defects in the laser welds such as precipitates and voids. These internal defects significantly influenced the local stress and strain at the time of residual stress and load stress. Figure 10a shows the transmission image observed by irradiating with synchrotron radiation X-rays from the side of the specimen. The white line shows the boundary between the molten pour and the base metal. The red frame denotes the region where the strain was measured. The molten portion contained multiple voids and cracks. However, they did not significantly affect the stress distribution during loading since the defects were small and well separated.

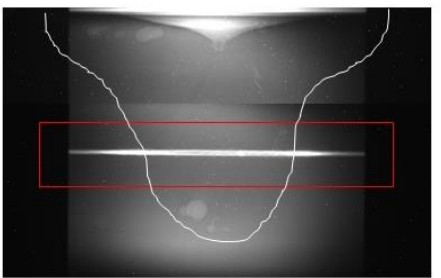 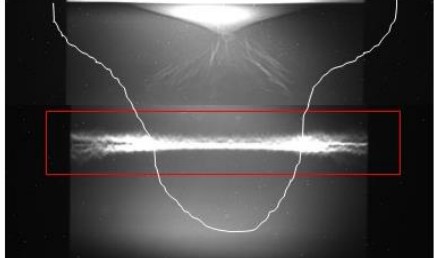

(**a**) Specimen discussed in this study        (**b**) Another specimen

**Figure 10.** Transmission imaging of the lap joint.

On the other hand, Figure 10b shows a transmission image of another lap joint weld prepared under the same conditions as those of Figure 10a. Compared with Figure 10a, the

contrast near the interface was bright, and cracks radially spread from the surface to the inside. The internal defects were random in the laser weld. Hence, the internal strain and stress associated with the shape of the internal defect differed.

Figure 11 shows the two-dimensional distribution of the lattice strain at 530 °C under a tensile load. Compared to Figure 5, a local strain near the interface was not generated, but the tensile strain in a part of the melt zone increased. It is presumed that the tensile load was not concentrated near the interface. Instead, it is thought that the stress was dispersed to the crack tips, which were not measured because the cracks were generated radially. Finite element analysis can introduce defects. In the future, if the positions of such defects are clarified three-dimensionally, more accurate strain predictions would be possible.

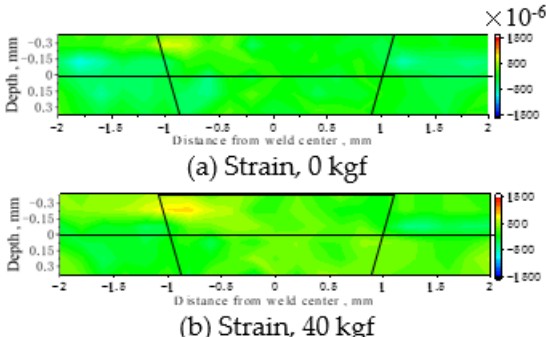

**Figure 11.** Two-dimensional distribution of lattice strain in the other specimen at 530 °C under a tensile load.

## 5. Conclusions

This study measured the internal strain distribution of laser lap joint PNC-FMS steels using high-energy synchrotron radiation X-ray diffraction. The results were compared with FEM analyses. The main findings are summarized as follows:

(1) As the tensile load increased, the local tensile and compressive strain increased near the interface, and the changes agreed well with the finite element analysis results.

(2) Complementary utilization of the internal defect observations by X-ray transmission imaging is crucial because the results depend on the defects generated by laser processing.

**Author Contributions:** F.K., T.Y., M.N. and T.O. performed laser welding experiments. T.S. and A.S. performed X-ray diffraction and transmission measurements. M.N. and T.O. performed FEM analyses. T.S., T.M. and T.O. designed and supervised the study. T.S. wrote the manuscript. All authors discussed the results and reviewed the manuscript. All authors have read and agreed to the published version of the manuscript.

**Funding:** This research received no external funding.

**Acknowledgments:** The authors would like to thank M. Takano at E&E Techno Service Co., Ltd. for his support in FEM analysis. The synchrotron radiation experiments were performed at the JAEA beamline BL22XU in SPring-8 with the approval of the Japan Synchrotron Radiation Research Institute (JASRI) (Proposal Nos. 2013B3721, 2014A3721, 2016A3772).

**Conflicts of Interest:** The authors declare no competing interest.

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
