# Peer review of "Internal Strain Distribution of Laser Lap Joints in Steel under Loading Studied by High-Energy Synchrotron Radiation X-rays"

_qubs, doi:10.3390/qubs5020017_

Round 1

Author Response

Thank you for the peer review.
The answer to the peer review is as shown in the attached file.

Reviewer 2 Report

This article measured the internal strain distribution (using high energy synchrotron radiation X-ray diffraction). The  method using neutrons shows a realize nondestructive measurements inside a material. The diffracted X-rays from the local region inside the sample were measured by a CdTe diode detector (there is no information about the detector “CdTe” ((line 78) you used yet, please fill out this data). Authors shows the excessive temperature changes induce the precipitates and voids.  However, they did not significantly affect the stress distribution. However, I have a note on eq. 1: this formula be written in the following form:  'delta'd/d=(d-dd)/d=sin ('teta')-sin('teta'0)/sin('teta') .  Accept after minor revision.

Author Response

(The authors gave the same response as above.)
